# Elevated Levels of Growth/Differentiation Factor-15 in the Aqueous Humor and Serum of Glaucoma Patients

**DOI:** 10.3390/jcm11030744

**Published:** 2022-01-29

**Authors:** Rupalatha Maddala, Leona T. Y. Ho, Shruthi Karnam, Iris Navarro, Anja Osterwald, Sandra S. Stinnett, Christoph Ullmer, Robin R. Vann, Pratap Challa, Ponugoti V. Rao

**Affiliations:** 1Department of Ophthalmology, Duke University School of Medicine, Durham, NC 27710, USA; rupa.maddala@duke.edu (R.M.); leona24ho@gmail.com (L.T.Y.H.); shruthi.karnam@berkeley.edu (S.K.); iris.navarro@duke.edu (I.N.); sandra.stinnett@duke.edu (S.S.S.); Robin.Vann@duke.edu (R.R.V.); 2Pharma Research and Early Development, Roche Innovation Center Basel, F. Hoffmann-La Roche Ltd., 4070 Basel, Switzerland; anja.osterwald@roche.com (A.O.); christoph.ullmer@roche.com (C.U.); 3Duke Eye Center, Duke University, Durham, NC 27710, USA

**Keywords:** glaucoma, aqueous humor, GDF15, serum, intraocular pressure

## Abstract

Dysregulated levels of growth/differentiation factor-15 (GDF15), a divergent member of the transforming growth factor-beta super family, have been found to be associated with the pathology of various diseases. In this study, we evaluated the levels of GDF15 in aqueous humor (AH) and serum samples derived from primary open-angle glaucoma (POAG) and age- and gender-matched non-glaucoma (cataract) patients to assess the plausible association between GDF15 and POAG. GDF15 levels were determined using an enzyme-linked immunosorbent assay, and data analysis was performed using the Wilcoxon rank sum test, or the Kruskal–Wallis test and linear regression. GDF15 levels in the AH (*n* = 105) of POAG patients were significantly elevated (by 7.4-fold) compared to cataract patients (*n* = 117). Serum samples obtained from a subgroup of POAG patients (*n* = 41) also showed a significant increase in GDF15 levels (by 50%) compared to cataract patients. GDF15 levels were elevated in male, female, African American, and Caucasian POAG patients. This study reveals a significant and marked elevation of GDF15 levels in the AH of POAG patients compared to non-glaucoma cataract control patients. Although serum GDF15 levels were also elevated in POAG patients, the magnitude of difference was much smaller relative to that found in the AH.

## 1. Introduction

Glaucoma, a group of optic neuropathies, is the second leading cause of irreversible blindness worldwide. Among the various characterized types of glaucoma, primary-open angle glaucoma (POAG) accounts for more than 70% [1,2]. Although the etiology of POAG is poorly understood, elevated intraocular pressure (IOP) due to impaired aqueous humor (AH) drainage through the trabecular pathway has been identified as a major risk factor for POAG [3,4]. Lowering of IOP has been demonstrated to slow down retinal ganglion cell death and vision loss in glaucoma patients, and is a mainstay of treatment for glaucoma [3,4,5]. Though several IOP-lowering drugs are currently available, many of them do not possess adequate efficacy to control IOP in glaucoma patients, with several posing considerable adverse effects [3]. Therefore, a better understanding of the etiology of ocular hypertension and impaired AH outflow through the trabecular pathway, which accounts for more than 80% of AH outflow in glaucoma patients, is necessary to enable the development of targeted and efficacious IOP-lowering drugs [4,6].

Dysregulated levels of various secretory proteins in the AH, including transforming growth factor-beta (TGF-β), endothelin-1, connective tissue growth factor, and lysophospholipid-producing enzyme autotaxin, have been shown to be associated with POAG and elevated IOP [7,8]. We recently reported that human trabecular meshwork cells express and secrete growth/differentiation factor 15 (GDF15), suggesting its plausible role in the regulation of IOP [9]. GDF15 is a distant member of the TGF-β superfamily, and is widely expressed in multiple mammalian tissues albeit in low concentrations [10,11]. GDF15 is often induced under stress and with aging [12], and is involved in the homeostasis of cell and tissue function [11]. Elevated GDF15 levels are linked to various pathological conditions, including inflammation, cardiovascular disease, cancer, obesity, and kidney disease [13,14,15]. GDF15 has thus been widely explored as a prognostic biomarker in several diseases [11,14].

Previous studies have reported significant elevations in AH and serum levels of GDF15 protein in small cohorts of POAG and pseudoexfoliation glaucoma patients, where the increased levels of AH GDF15 exhibited a positive correlation with disease severity [16,17,18]. The findings from these studies were inconsistent, however [18], and did not include an evaluation of GDF-15 in AH and serum derived from the same cohort of patients [16,19]. Therefore, to undertake an independent evaluation of the status of GDF15 levels in the AH and serum from the same population of glaucoma patients, and to understand the plausible role of GDF15 in the pathobiology of ocular hypertension, we determined the levels of GDF15 in AH and serum samples derived from POAG and age- and gender-matched non-glaucoma (cataract) patients. The findings of this study reveal elevated levels of GDF15 in the AH, as well as the serum, of POAG patients, indicating the plausible role for GDF15 in the pathobiology of ocular hypertension.

## 2. Materials and Methods

Human Subjects: Research involving the collection of human AH and blood samples was approved by the Institutional Review Board (IRB; Protocol No. Pro00093311)/Ethics Committee of Duke University Medical Center, in compliance with Health Insurance Portability and Accountability Act guidelines, and the tenets of the Declaration of Helsinki. Written informed consent was obtained from patients prior to the collection of AH and blood samples. All samples analyzed in the study were obtained from patients who underwent cataract or glaucoma surgeries at the academic Duke University Eye Center. Only one eye per patient was enrolled for glaucoma and non-glaucoma subjects. Samples were collected starting from the beginning of 2017 to the end of 2020.

Clinical Assessment: Patients underwent a review of medical history, measurement of best-corrected visual acuity and refraction, slit lamp biomicroscopy, gonioscopy, and Goldmann applanation tonometry (Haag-Streit Diagnostics, Essex, UK). In glaucoma patients, disc and red-free fundus photography, gonioscopy, and optical coherence tomography (OCT; Heidelberg Engineering Inc., Franklin, MA, USA) were performed. POAG was defined as the presence of glaucomatous optic neuropathy associated with typical reproducible glaucomatous visual field defects without any other ocular disease or conditions that might elevate IOP, and an open angle on gonioscopy. A glaucomatous visual field change was defined either as: (1) an outside normal limit result on the glaucoma hemifield test; (2) three abnormal points with a <5% probability of being normal, including one with a probability <1% by pattern deviation; or (3) a pattern standard deviation of 5% if the visual field was otherwise full, as confirmed on two consecutive tests. Disease severity of glaucoma, including mild, moderate, and severe classification, was based on the ICD-10 coding of each patient. Pigment dispersion syndrome and psuedoexfoliation syndrome cases were excluded. In the cataract and POAG patients, the rate of comorbidities (systemic diseases), medications (including cholesterol-, glucose-, and hypertension-lowering drugs; atropine, multivitamin supplement, calcium supplement, Tylenol, gabapentin, and Flonase), and smoking were found to be very close. Use of sildenafil was identified in some of the cataract patients, but was not significant. As anticipated, most of the POAG patients were on one or more IOP-lowering medications prior to glaucoma surgery.

Collection of aqueous humor and blood samples: AH samples were collected at the initiation of cataract or glaucoma surgery. A tuberculin syringe with a 30-gauge needle was inserted into the anterior chamber through a limbal paracentesis tract at the start of the surgery, and approximately 40–100 μL of AH was slowly aspirated. The AH samples were transferred from the syringe to a 1.5 mL Eppendorf tube, and centrifuged at 1000× *g* for 10 min at 4 °C. The supernatant obtained from the AH samples was collected and stored at −80 °C until further use.

Blood samples from the patients were collected by venipuncture by the triage nurse using a BD Vacutainer push button blood collection set, and the blood was carefully dispensed into serum blood collection tubes from Becton, Dickinson, and Company, Franklin Lakes, NJ. Tubes containing blood samples were inverted gently five times, rested at room temperature for 30 min, and centrifuged for 15 min at 1000× *g* using a benchtop Eppendorf centrifuge at room temperature. Serum samples were collected into bio-safe tubes, and stored at −80 °C until further use.

Detection of GDF15: A human GDF15 enzyme-linked immunosorbent assay kit (Human GDF15 DuoSet ELISA, R&D Systems, Inc., Minneapolis, MN, USA) was used to determine the levels of GDF15 in both AH and serum samples. Analysis was performed in a masked manner, and per the manufacturer’s protocol, which included appropriate standards and background controls, using a SpectraMax M3 plate reader (Molecular Devices, San Jose, CA, USA). Ten microliters of AH was used in duplicate for each sample analyzed. Serum samples were diluted 10-fold with diluent buffer provided in the kit, and 10 µL of the diluted sample was used in duplicate per sample. Results for both AH and serum samples were expressed as picograms of GDF15/milliliter (pg/mL).

Statistical Analysis: The significance of differences in continuous variables between categories of disease (POAG and cataract), diagnosis (cataract, mild POAG, moderate POAG, severe POAG), race (Asian, African American, Caucasian), and gender were assessed using the Wilcoxon rank sum test or the Kruskal–Wallis test. Relationships between continuous variables were assessed using linear regression. The data analysis was performed using SAS/STAT software, Version 9.4 of the SAS System for Windows (Copyright © 2022–2012 SAS Institute Inc.).

## 3. Results

### 3.1. Elevated Levels of GDF15 in Both AH and Serum Samples of POAG Patients

In this study, we aimed to investigate whether dysregulated AH and serum GDF15 levels relate to POAG. To address this goal, we initially compared the levels of GDF15 in AH and serum samples from the same cohort of POAG patients (*n* = 40 and 41, respectively) with age- and gender-matched controls (non-glaucomatous, cataract patients, *n* = 32). Although this cohort was matched for gender and age, race was not matched between the glaucoma and cataract patients in the study (Appendix A describes the demography details of human patients included in the study). The GDF15 levels in AH and serum samples from POAG patients were significantly elevated (*p* < 0.001 and *p* < 0.011, respectively) by >9 fold and 1.5-fold, respectively, compared to the respective samples derived from non-glaucoma (cataract) patients based on the Wilcoxon rank sum test of difference between medians (Figure 1, Appendix A). AH GDF15 levels were significantly (*p* < 0.001) elevated in both male (by 11.6-fold) and female (by 5.4-fold) POAG patients compared to cataract patients (Appendix A). In the case of serum samples (Appendix A), although there was an increase (median values: 2228.0 pg/mL, *n* = 19) in male POAG patients compared to male cataract patients (1850.0 pg/mL, *n* = 19), the difference did not achieve statistical significance (*p* < 0.148). Serum GDF15 levels in female POAG patients (*n* = 22), however, were significantly (*p* < 0.015) elevated (by 64%) compared to female cataract patients (*n* = 13).

### 3.2. Robust Elevation of GDF15 Levels in the Aqueous Humor of a Large Cohort of POAG Patients

We then expanded the scope of our investigation to include analysis of GDF15 in AH samples from additional POAG and cataract patients. Analysis of GDF15 was performed as described above, with the data set comprised of a total of 105 POAG AH samples and 117 non-glaucoma cataract AH samples derived from gender- and age-matched, but not race-matched, patient populations (Appendix A describes the demography details for all of the POAG patients of this study). Median GDF15 levels in the AH from this large cohort of subjects were significantly (*p* < 0.001) and robustly (by 7.4-fold) elevated in the POAG group relative to the cataract group (Figure 2, Appendix A). The coefficient of variation for GDF15 in POAG is 104.9. In this large cohort of POAG patients, males (*n* = 52) and females (*n* = 53), and African Americans (*n* = 53) and Caucasians (*n* = 50), all showed significantly increased median levels of AH GDF15 (*p* < 0.001) compared to their respective non-glaucoma (cataract) controls (males, *n* = 53; females, *n* = 64; African Americans, *n* = 16; Caucasians, *n* = 94). These results are summarized in Table 1 and Table 2. Within the POAG patient group, there was no gender or racial (African American versus Caucasians) difference in the AH GDF15 levels (data not shown). Interestingly, although there was no racial difference in the AH GDF15 levels within the cataract group, the male subjects (*n* = 53) have significantly (*p* < 0.046) higher median levels (by 33%) of AH GDF15 compared to female subjects (*n* = 64, Appendix A). This gender difference in the levels of GDF15 was also consistent in the serum samples derived from the cataract subjects (Appendix A). Serum samples derived from male cataract (*n* = 19) subjects have significantly (*p* < 0.025) higher (by 42%) median levels of GDF15 compared to the female cataract subjects (*n* = 13).

### 3.3. Aqueous Humor GDF15 Levels, IOP, and Age Relationships in POAG Patients

Intraocular pressure was significantly higher (*p* < 0.001) in POAG patients (*n* = 105; 20.79 ± 6.974 (mean ± SD) compared to non-glaucoma cataract patients (*n* = 43; 14.97 ± 2.923; Appendix A). The coefficient of variation for IOP in POAG is 33.5. However, AH and serum GDF15 and IOP did not reveal a significant relationship in POAG patients (Appendix A, respectively). Similarly, there was no significant relationship between the AH GDF15 levels and age, and between the serum GDF15 and age in POAG patients (Appendix A, respectively).

### 3.4. Relationship between AH GDF15 and POAG Disease Severity

To explore a plausible relationship between the levels of AH GDF15 and disease severity of glaucoma, we sub-classified glaucoma patients from the smaller cohort described above (Appendix A) into mild (*n* = 10), moderate (*n* = 8), and severe (*n* = 20) POAG subgroups, based on the ICD-10 coding of each patient. The mean deviation was taken from the last visual field prior to surgery. All visual fields were within 3 months of surgery. The median deviation was significantly different between the mild POAG and severe POAG patients, and between the moderate POAG and severe POAG subgroup, but not between mild POAG and moderate POAG subgroups (Figure 3A). As expected, cup-to-disk ratio was significantly higher in mild, moderate, and severe POAG compared to that of cataract patients (*n* = 43). The cup-to-disk ratio was significantly higher in moderate POAG compared to mild POAG, and in severe POAG compared to mild POAG patients (Figure 3B). AH GDF15 levels were significantly elevated in mild, moderate, and severe POAG patients compared to cataract patients. However, although there was an increase in GDF15 levels in the severe POAG patient subgroup relative to mild POAG patients, there was no statistical difference in AH GDF15 levels between the mild and moderate POAG, and between the mild and severe POAG patients (Figure 3C). Additionally, we examined for an association between the serum GDF15 levels and severity of POAG. Though there was a significant increase in the serum GDF15 levels in moderate (*n* = 8) and severe (*n* = 21) POAG patients compared to cataract patients (*n* = 32), there was no difference between the mild POAG (*n* = 10) and severe POAG patients (Figure 3D). Interestingly, serum GDF15 levels were found to be significantly higher in the moderate POAG patients compared to mild POAG patients (Figure 3D). Overall, although there is an increase in the levels of both AH and serum GDF15 levels in severe POAG patients compared to mild POAG patients, there was no statistical difference between these two groups in the small cohort of glaucoma patients examined.

## 4. Discussion

GDF15 is a well-characterized secretory stress cytokine that possesses several physiological activities [11,13]. Our previous study identified GDF15, a divergent member of the TGF-β superfamily of growth factors, as a common constituent of the extracellular matrix secreted by trabecular meshwork cells [9]. The results of this study revealed a robust increase in AH GDF15 levels, with a moderate and significant increase in serum GDF15 levels of POAG patients relative to the corresponding samples from control (non-glaucomatous cataract) patients, suggesting a potential association between GDF15 and POAG.

GDF15, which was also identified as macrophage inhibitory cytokine-1 and nonsteroidal anti-inflammatory drug activated gene-1, has gained increased attention because of its role in several diseases [11,13]. Increased GDF15 levels have been reported under various disease conditions, including cancer, cardiovascular disease, liver and kidney diseases, and with inflammation, age, smoking, stress, and tissue injury [11,14,15,20]. Although it was initially thought to act through the TGF-β receptors and possess TGF-β-like biological activities, recent studies have revealed that GDF15 binds to the GDNF family receptor α–like (GFRAL) orphan receptor with high affinity, with GFRAL playing a crucial role in GDF15-induced weight loss and decreased food intake [21,22,23]. However, GFRAL is reported to be expressed only in a distinct region of the brain known as the postrema [10,23]. Therefore, it is presumed that the widely distributed GDF15 may also act through an alternative, but yet to be characterized, molecular pathway(s), and to be involved in different physiological activities [13].

Interestingly, in the eye, GDF15 has been shown to promote retinal ganglion cell (RGC) differentiation [24], and neuroprotection of RGC under optic nerve crush injury [25], with GDF15 expression and secretion by RGCs being increased under tissue injury [16]. Moreover, GDF15 was found to be elevated in the RGC and AH in different glaucoma experimental models [16], wherein a good correlation has been documented between RGC loss and increased expression and protein levels of GDF15 [16]. Importantly, elevated AH GDF15 has also been shown to correlate with disease severity (based on the median deviation of visual field loss) in POAG and pseudoexfoliation glaucoma patients [16,17]. Based on these findings, GDF15 has been suggested to serve as a molecular marker for glaucomatous neurodegeneration [16]. However, these latter conclusions were drawn mainly based on findings from one laboratory study [16,17]. Moreover, although serum derived from both POAG and pseudoexfoliation glaucoma patients has been reported to contain elevated levels of GDF15 [19], a recent study using plasma derived from both POAG and pseudoexfoliation glaucoma patients showed no difference in GDF15 levels compared to non-glaucoma (cataract) patients [18]. The reason for the noted discrepancy between levels of GDF15 in serum and plasma derived from glaucoma patients is not clear. However, in the above referenced studies, there was no evaluation of GDF15 levels in the AH and serum derived from the same cohort of patients.

In this study, using serum and AH from the same cohort of patients, we found significantly elevated levels of GDF15 in POAG patients relative to cataract patients, with a more robust difference in AH GDF15 levels compared to serum GDF15 levels. A total of 105 POAG and 117 cataract patient derived samples were utilized for the analysis of AH GDF15 levels. In this relatively large cohort of POAG patients, the AH GDF15 levels were robustly elevated (by 7.4-fold) relative to the respective age- and gender-matched non-glaucoma cataract patient subgroup. This increase in AH GDF15 levels was consistent in males, females, African American, and Caucasian POAG patients. However, when the subset of POAG patients was categorized into mild, moderate, and severe POAG subgroups based on the median deviation of visual field loss, we did not find a definitive correlation between the levels of AH or serum GDF15 and disease severity of glaucoma, although there was an increasing trend in GDF15 levels in AH and serum with an increasing disease severity of glaucoma. The sample size in our cohort of glaucoma patients was very close to that described by Lin et al. [17], and Ban et al. [16], in which they found a positive correlation between the levels of GDF15 of AH and the disease severity of glaucoma.

Interestingly, consistent with previous reports [18,26], AH and serum samples from male cataract patients contained significantly higher levels of GDF15 relative to female patients, suggesting a definitive gender difference in GDF15 levels. Coincidently, the gender prevalence of open angle glaucoma is mixed, but some studies have reported higher rates in males compared to females [1,27,28]. However, within POAG patients, there was no statistical difference in AH GDF15 levels between male and female patients.

## 5. Limitations

We want to recognize the limitations of this study in regards to a lack of a definitive correlation between the levels of AH GDF15 and the disease severity of glaucoma. The sample size in our study was one limitation for establishing this correlation. Additionally, though this study matched the gender and age of subgroups, race was not matched between the POAG and cataract (non-glaucoma) subgroups, and race and ethnicity have been found to influence the incidence of glaucoma [1]. Additionally, this study did not reveal a correlation between the levels of AH or serum GDF15 and IOP in POAG patients. Two reasons could likely account for this observation, with the first being that all POAG patients were already on IOP-lowering medications as part of their glaucoma therapy; as a result of which, their IOP readings would be lower. Since we did not have access to all of the patients’ pre-treatment IOPs, this possibility could not be verified. The second reason is that we do not know the implications of prior eye surgeries for GDF15 levels, since some POAG patients had previously undergone cataract removal and glaucoma procedures, and corneal, or retinal intraocular surgery.

## 6. Conclusions

This study reveals increased levels of GDF15 in both the AH and serum of POAG patients compared to age- and gender-matched non-glaucoma cataract patients, suggesting a strong association between elevated levels of GDF15 and POAG.

## Figures and Tables

**Figure 1 jcm-11-00744-f001:**
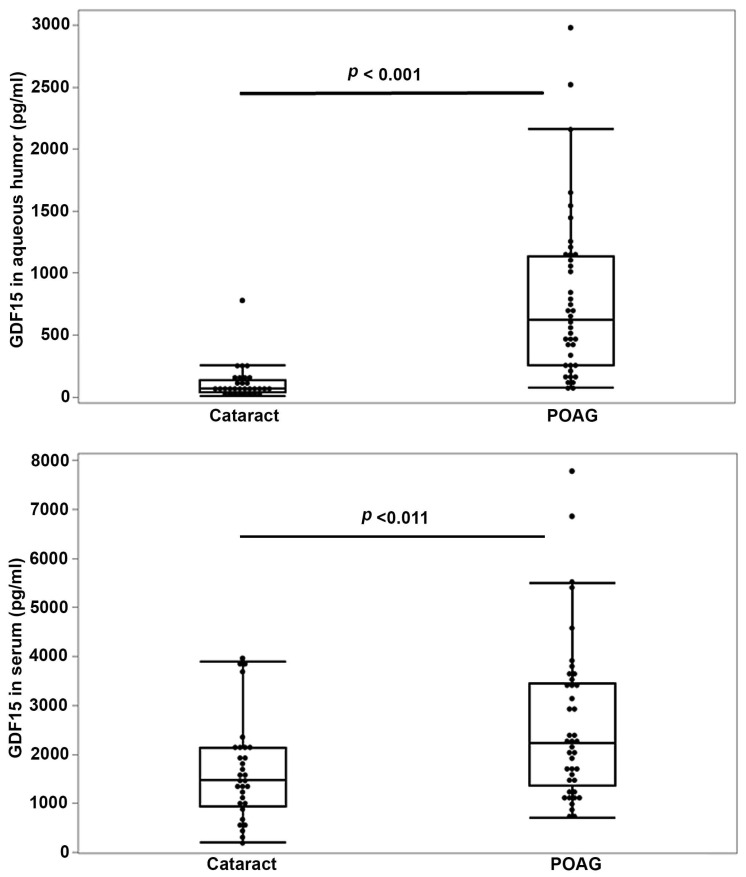
Elevated levels of GDF15 in aqueous humor and serum samples of POAG patients. Aqueous humor (*n* = 40) and serum (*n* = 41) samples derived from POAG patients revealed significantly elevated levels of growth/differentiation factor-15 (GDF15) (by >9-fold and 50%, respectively) compared to age- and gender-matched cataract patient samples (*n* = 32). The box and whisker plots represent median and the interquartile range in the distribution. *p*-values were based on the Wilcoxon rank sum test of difference between medians. Abbreviations: GDF15, growth/differentiation factor-15; POAG, primary open-angle glaucoma.

**Figure 2 jcm-11-00744-f002:**
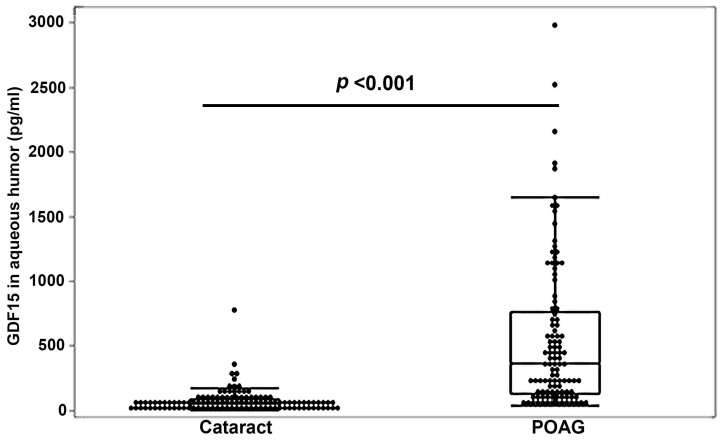
Elevated levels of GDF15 in aqueous humor samples from POAG patients. Aqueous humor samples derived from POAG patients (*n* = 105) showed significantly increased levels of GDF15 (by 7.4-fold) compared to age- and gender-matched non-glaucoma cataract patient samples (*n* = 117). The box and whisker plots represent median and the interquartile range in the distribution. *p*-values were based on Wilcoxon rank sum test of difference between medians. Abbreviations: GDF15, growth/differentiation factor-15; POAG, primary open-angle glaucoma.

**Figure 3 jcm-11-00744-f003:**
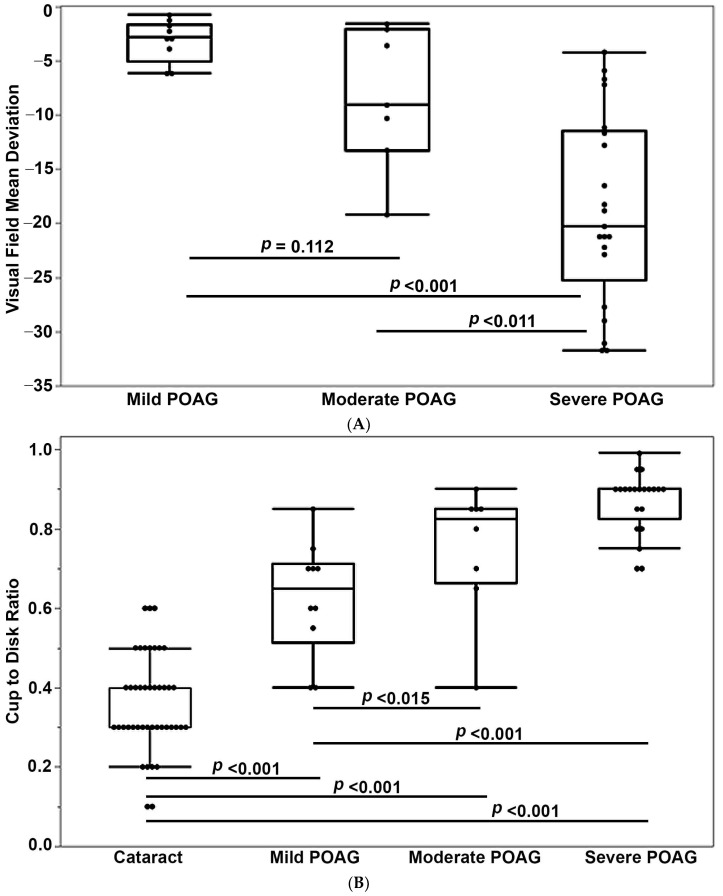
Relationship between aqueous humor and serum GDF15 levels and disease severity of POAG. To determine whether elevated levels of GDF15 either in the AH or serum reveal a positive association with disease severity of glaucoma, POAG patients from a small cohort study were divided into mild, moderate, and severe glaucoma based on median deviation of visual field loss and GDF15 levels from the categories analyzed. (**A**) Overall, there was a significant difference in median deviation of visual field loss among the three groups, based on the Kruskal–Wallis test (*p* < 0.001). Though the median deviation was not found to be significant between mild and moderate POAG, it was significant between mild to severe, and moderate to severe POAG based on the Wilcoxon rank sum test. (**B**) Cup-to-disk ratio showed a significant difference between cataract patients versus mild, moderate, and severe POAG, and between mild to moderate, and mild to severe POAG based on the Wilcoxon rank sum test. (**C**) AH GDF15 levels were significantly elevated between cataract and mild, moderate, and severe POAG, but not different between mild and moderate, and mild and severe POAG, based on the Wilcoxon rank sum test. (**D**) Serum GDF15 levels were also significantly elevated in moderate and severe POAG samples compared to cataract samples, but not different between mild and severe POAG, although there was a significant difference between mild and moderate POAG based on the Wilcoxon rank sum test. In all panels, the box and whisker plots represent median and the interquartile range in the distribution. Abbreviations: GDF15, growth/differentiation factor-15; POAG, primary open-angle glaucoma.

**Table 1 jcm-11-00744-t001:** GDF15 levels in the aqueous humor of male and female POAG and cataract patients.

	Gender		Cataract	POAG	*p*-Value *
GDF-15 (AH)	Female	*n*	64	53	
		Mean (SD)	60.41 (65.33)	543.55 (664.30)	
		Min, Median, Max	0.3, 39.0, 357.1	36.1, 246.0, 2978.0	<0.001
GDF-15 (AH)	Male	*n*	53	52	
		Mean (SD)	80.07 (108.36)	556.95 (478.73)	
		Min, Median, Max	1.5, 52.0, 776.0	48.0, 438.6, 1868.2	<0.001

* *p*-value based on Wilcoxon rank sum test of difference between medians.

**Table 2 jcm-11-00744-t002:** GDF15 levels in the aqueous humor of Caucasian and African American POAG and cataract patients.

Variable	Race	Statistic	Cataract	POAG	*p*-Value *
GDF-15 (AH)	African	*n*	16	53	
	American	Mean (SD)	54.90 (43.12)	515.41 (561.68)	
		Min, Median, Max	2.0, 49.0, 146.0	44.7, 225.0, 2156.0	<0.001
	Caucasian	*n*	94	50	
		Mean (SD)	74.26 (95.43)	571.55 (591.09)	
		Min, Median, Max	0.3, 51.0, 776.0	36.1, 419.5, 2978.0	<0.001

* *p*-value based on Wilcoxon rank sum of difference between medians.

## Data Availability

Authors will make all data available upon request.

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
