# Peer review of "Elevated Levels of Growth/Differentiation Factor-15 in the Aqueous Humor and Serum of Glaucoma Patients"

_jcm, 2022, doi:10.3390/jcm11030744_

Round 1

Reviewer 1 Report

Maddala et al. investigated the levels of GDF15 in AH and serum samples pf POAG patients.

The authors analyzed both the smaller cohort and large cohort. I recommend the authors should analyze only the large cohort.

When the relationship between the levels of AH GDF15 and diseases severity of glaucoma, the authors should analyze the large cohort (105 POAG patients).

I do not understand well why the authors showed the data of both the smaller cohort and large cohort. If serum sample obtained from only the smaller cohort of POAG patients, the authors should analyze only the small cohort, not show the data of the large cohort. It is confusing.

Smoking may increase GDF15 levels. However, the authors did not pay attention in this study sample.

Author Response

Response to Reviewer 1:

We thank the reviewer for his/her thorough review and helpful suggestions for strengthening the conclusions of the study and quality of manuscript.

We reviewed the Methods section again to make sure that we included all the required details for the described analyses including the patient groups, demography, sample collection, ELISA and statistical software used. We also revised some of the text under the study Limitations section (marked in red). 

Comment 1: The authors analyzed both the smaller cohort and large cohort. I recommend the authors should analyze only the large cohort.

Response: We respect the reviewer’s suggestion however, as we discussed in our manuscript, there were no prior studies on the levels of GDF15 in aqueous humor (AH) and serum derived from the same set of glaucoma patients. Moreover, there was only one prior study on the levels of GDF15 in the AH of POAG patients. Therefore, we had two objectives for this study: Objective one: Compare the levels of GDF15 in both AH and serum derived from the same set of POAG and age and gender matched non-glaucoma (cataract) patients in a smaller group since not all patients consent for blood collection, and the latter aspect is time consuming and requires more resources. Objective two: Evaluate the levels of GDF15 in AH in higher number of POAG patients derived from a different patient study group (from the Duke Eye Center) compared to the previously published data from the Washington University to gain more insight in to the changes in the GDF15 levels regarding its racial, gender and age differences between POAG and non-POAG patients. It is well recognized that POAG is more prevalent in African Americans compared to Caucasians. In the smaller group of POAG patients in which we determined the levels of GDF-15 in both AH and serum, we did not have enough sample size for evaluating the racial differences in GDF15. We believe that our analyses are very informative and comprehensive. For these reasons, we decide to include all the data that we described for both serum and AH specimens derived from the POAG patients.

Comment 2:  When the relationship between the levels of AH GDF15 and diseases severity of glaucoma, the authors should analyze the large cohort (105 POAG patients).    

Response: Although we wished to gain a comprehensive insight into GDF15 regarding its possible association with disease severity of POAG, unfortunately, we did not have visual field data for all the patients.

Comment 3: I do not understand well why the authors showed the data of both the smaller cohort and large cohort. If serum sample obtained from only the smaller cohort of POAG patients, the authors should analyze only the small cohort, not show the data of the large cohort. It is confusing.

Response: Please see our response to Comment 1. As commented and complimented by reviewer two, we feel that the data provided for both serum and AH samples, in both smaller and larger patient groups have very important information related to the objectives described under response 1.  

Comment 4: Smoking may increase GDF15 levels. However, the authors did not pay attention in this study sample.

Response: Although smoking has been found to influence GDF15 levels, in our patient population, there were very negligible number of smokers and was not different between POAG and cataract groups described in this study. We provided these details in the manuscript.  

Reviewer 2 Report

Reviewer compliments authors for conducting this well designed and novel study. Growth differentiation factor 15 (GDF-15) also known as serum macrophage inhibitory cytocine-1 (MIC-1), is a member of the TGF-β superfamily and has been identified as molecular marker associated with inflammation and cellular injury especially in cardiomyocytes as a response to oxidative stress , proinflammatory cytokines, ischaemia or mechanical stress and stretch. In addition to cardiomyocytes, GDF-15   has been found to have association with extra cardiac cells like adipocytes, pulmonary endothelial cells , macrophages ,erythropoiesis and smooth muscles cells as well as in cases of pancreatic cancer.

Reviewer could identified two recent studies. Wouter H G Hubens etall concluded that Plasma GDF-15 concentration is not elevated in open-angle glaucoma(1).Where as Jonathan B. Lin etall  identified elevated  GDF15 in serum and in aqueous humor as a molecular marker of retinal ganglion cell stress in rodent models of glaucoma as a possible risk factor for glaucoma progression. Hence more scientific studies that to having larger series and detailed studies of variable are essentially needed to establish a well drawn conclusion .

References

1.Wouter H G Hubens , Mariëlle T Kievit , Tos T J M Berendschot , Irenaeus F M de Coo , Hubert J M Smeets , Carroll A B Webers , Theo G M F Gorgels . Plasma GDF-15 concentration is not elevated in open-angle glaucoma. PMID: 34048486 ,PMCID: PMC8162581,DOI: 10.1371/journal.pone.0252630

 2.Jonathan B. Lin,  Arsham Sheybani,  Andrea Santeford,  and Rajendra S. Apte, Longitudinal Growth Differentiation Factor 15 (GDF15) and Long-term Intraocular Pressure Fluctuation in Glaucoma: A Pilot Study. J Ophthalmic Vis Res. 2021 Jan-Mar; 16(1): 21–27. PMCID: PMC7841272.PMID: 33520124.

Author Response

We thank the reviewer for a very positive assessment of our study.

Round 2

Reviewer 1 Report

No comments.